# Detecting Targets above the Earth's Surface Using GNSS-R Delay Doppler Maps: Results from TDS-1

**Changjiang Hu** [1,*], **Craig Benson** [1], **Hyuk Park** [2], **Adriano Camps** [3], **Li Qiao** [1] **and Chris Rizos** [4]

1. School of Engineering and Information Technology, University of New South Wales, 2612 Canberra, Australia; c.benson@adfa.edu.au (C.B.); l.qiao@adfa.edu.au (L.Q.)
2. Department of Physics, Univeristat Politecnica de Catalunya, 08034 Barcelona, Spain; park.hyuk@tsc.upc.edu
3. Department of Signal Theory and Communications, Universitat Politecnica de Caltalunya and IEEC/UPC, 08034 Barcelona, Spain; camps@tsc.upc.edu
4. School of Civil and Environmental Engineering, University of New South Wales, 2052 Sydney, Australia; c.rizos@unsw.edu.au
* Correspondence: changjiang.hu@student.adfa.edu.au; Tel.: +61-0415-365-889

**Abstract:** Global Navigation Satellite System (GNSS) reflected signals can be used to remotely sense the Earth's surface, known as GNSS reflectometry (GNSS-R). The GNSS-R technique has been applied to numerous areas, such as the retrieval of wind speed, and the detection of Earth surface objects. This work proposes a new application of GNSS-R, namely to detect objects above the Earth's surface, such as low Earth orbit (LEO) satellites. To discuss its feasibility, 14 delay Doppler maps (DDMs) are first presented which contain unusually bright reflected signals as delays shorter than the specular reflection point over the Earth's surface. Then, seven possible causes of these anomalies are analysed, reaching the conclusion that the anomalies are likely due to the signals being reflected from objects above the Earth's surface. Next, the positions of the objects are calculated using the delay and Doppler information, and an appropriate geometry assumption. After that, suspect satellite objects are searched in the satellite database from Union of Concerned Scientists (UCS). Finally, three objects have been found to match the delay and Doppler conditions. In the absence of other reasons for these anomalies, GNSS-R could potentially be used to detect some objects above the Earth's surface.

**Keywords:** GNSS reflectometry; delay Doppler map; target detection

## 1. Introduction

The use of Global Navigation Satellite System reflectometry (GNSS-R) to remotely sense the Earth's surface was originally proposed more than three decades ago [1]. GNSS-R works as a bistatic radar with receivers having two antennas: one up-looking antenna receiving the direct signals, and one down-looking antenna receiving the reflected signals. GNSS-R can be ground-based, airborne or spaceborne. During the first two decades of GNSS-R, a range of low-altitude (ground or airborne) campaigns were carried out to verify its feasibility [2–4]. In recent years, spaceborne GNSS-R has attracted increasing attention for its advantages of global data collection and data coverage capability. The first mission equipped with a GNSS-R payload was UK Disaster Monitoring Constellation (UK-DMC) which was in operation from 2003 to 2011 [5]. The follow-on mission of the UK-DMC was TechDemoSat-1 (TDS-1), which ceased operations in 2018 [6]. NASA's Cyclone GNSS (CYGNSS) mission is now operating [7]. Other missions being planned or in preparation can be found in [8–11].

GNSS-R applications known to date are broadly divided into two categories: retrieval of surface parameters, and object detection. The former can be further divided to ocean and land parameters

retrieval. GNSS-R ocean altimetry was first proposed in 1993 [12], it could achieve good performance when the reflecting surface is smooth, such as lake surfaces [13], and ice [14,15]. It is still challenging to obtain accurate results with open ocean reflections. Using GNSS reflected signals to determine land moisture was proposed in 2000 [16], and experimental results have been reported in [17] where the moisture of bare and vegetated soil is estimated using GNSS interferometric signals. In 1998 the effect of ocean roughness on reflected GNSS signals was detected [18], and sea roughness was retrieved using a model to fit the observed delay Doppler maps (DDMs) in an aircraft experiment in 2004 [19]. Wind speed retrieval using GNSS reflected signals was theoretically studied in [20], and in recent years ocean wind speed was retrieved using space-based GNSS-R data, such as [21] using TDS-1 data. In [21] precision around 2.2 m/s was achieved, which demonstrates the potential of ocean wind speed using GNSS-R. CYGNSS is another mission dedicated to study wind speed. GNSS-R can retrieve other parameters, such as forest biomass [22], snow depth [23], and ionospheric delay and scintillation [24]. For object detection, objects on the Earth's surface that can be detected by GNSS-R include sea ice, sea surface targets, oil slicks, and tsunamis, etc. The detection of sea ice, sea surface target and oil slicks is based on the reflection coefficient differences between the targets and the reflection bodies around them. For example, the signals reflected from ice are very strong which exhibits different features from ocean reflections in DDMs. An example of using TDS-1 DDMs to detect sea ice over Arctic and Antarctic regions can be found in [25]. A technique using two separate antennas to detect oil slicks was presented in [26]. Different from the detection of the sea ice, sea surface targets and oil slicks, tsunami detection uses the high spatial resolution of GNSS-R altimetric measurements [27]. The target detection mentioned above utilises space-based GNSS-R which has the advantage of short revisit time compared to traditional synthetic aperture radar and optical systems [28]. In 2018, it was found that GNSS-R can be used to detect traffic flow [29], which is based on phase difference information of the reflected signals. In [29], a ground-based experiment is employed, and if the space-based GNSS-R is feasible to detect the traffic flow has not been reported yet.

Different from the aforementioned applications, this paper proposes a new application of GNSS-R. The previous studies usually use signals reflected from the Earth's surface. However, this study concludes that the GNSS-R technique may be able to detect objects above the Earth's surface, such as spacecraft. Traditionally such detection has been done using ground-based radar technologies [30,31], which are regional systems. The traditional techniques have limited detection areas. However, GNSS-R has several advantages over the traditional techniques, such as global coverage, large volume of data, and short revisit time. Detecting targets above the Earth's surface not only extends the applications of GNSS-R, but also provides an alternative approach to solve this issue.

This work presents experimental evidence and demonstrates the process to verify the feasibility of target detection. Firstly, Section 2 of the study presents 14 DDMs from the UK TDS-1 mission which contain "anomalous" reflections. In Section 3 five probable causes for the anomaly are analysed, reaching the finding that the anomalous reflections are likely from objects above the Earth's surface based on their path delays. Section 4 describes the determination of an object's position using delay and Doppler frequency information. In order to find the suspected objects, in Section 5 the satellite database from the Union of Concerned Scientists (UCS) [32] has been searched, and three satellites satisfying the delay and Doppler conditions have been identified. Section 6 includes conclusions and remarks.

## 2. TDS-1 Satellite GNSS-R Data

The TDS-1 satellite was launched in 2014, and ceased its operations at the end of 2018. Although its initial purpose was to study wind speed, its GNSS-R data has been studied for many other applications, such as altimetry [14,15,33], ice detection [25], and sea target detection [28], to name a few. TDS-1 provides three kinds of data products according to processing level: L0 raw sampling data, L1 DDMs data, and L2 wind speed product [34]. A comprehensive introduction to the TDS-1 mission and its data products can be found in [6,34].

The data used in this paper are the L1 DDMs which are generated on-board by 1 ms coherent integration, and 1000 ms non-coherent integration. Figure 1 is a typical TDS-1 DDM over the ocean. The TDS-1 DDMs have 20 pixels in the Doppler axis, and 128 pixels in the delay axis. The resolutions (one pixel in Doppler and delay) are 500 Hz in Doppler, and 244 ns in delay. The Doppler of column 11 is zero which compensates the Doppler frequency of specular reflection. The specular point through which the reflected signal has the shortest path length is around the black dot marked in Figure 1. The "horseshoe" pattern of Figure 1 indicates reflected signals away from the specular reflection point, and the blue background above the black dot are noisy pixels without a return [28].

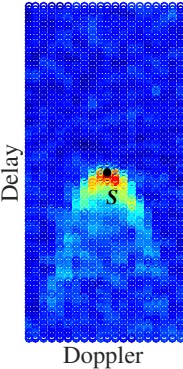

**Figure 1.** An example of delay Doppler map of TDS-1 satellite. *S* is specular point.

Fourteen anomalous DDMs that are used in this study are shown in Figure 2. They were all collected on 14 and 15 September 2018, and they are found from a total of 224,805 DDMs. Each DDM in Figure 2 is referred to as an "event". The location of the specular point of each event is marked by a red star in Figure 3. The reflecting bodies of the 14 events include water and land. The water reflections usually produce a clear "horseshoe" pattern in DDMs, such as Events 11 and 14 in Figure 2. Land reflections usually do not exhibit such a clear "horseshoe" pattern, such as Events 5 and 13 in Figure 2. The anomalies discussed in this paper are the bright points in the red circles of Figure 2. Table 1 gives the values of two important parameters (elevation angle and "*DirectSignalInDDM*") of the 14 DDMs, which are used in following analyses. The "*DirectSignalInDDM*" is a system parameter of the TDS-1 mission which indicates if the DDM contains direct signals. More details about this parameter can be found in [34]. In addition, the locations of the 14 DDMs in TDS-1 dataset are listed in Table 1 by track ID and observation time. For example, the DDM of Event 1 is in Group 05, Folder H18 of the metadata collected on 14 September 2018, and the observation time is 15:19:26.999 of the corresponding date. The third column of Table 1 gives the column number of the bright point in the DDM.

**Table 1.** Information of the 14 DDMs.

|  | Elevation Angle | Column No. of Bright Point | *"DirectSignal InDDM"* | Track ID in Dataset | Observation Time |
|---|---|---|---|---|---|
| Event 1 | 71° | 11 | False | 20180914H18Group05 | 15-19-26.999 |
| Event 2 | 72° | 11 | False | 20180914H18Group05 | 15-20-07.999 |
| Event 3 | 73° | 11 | False | 20180914H18Group05 | 15-20-58.999 |
| Event 4 | 64° | 17 | False | 20180914H18Group08 | 15-33-15.999 |
| Event 5 | 85° | 12 | False | 20180914H18Group51 | 18-51-51.999 |
| Event 6 | 59° | 8 | False | 20180914H18Group82 | 20-21-29.999 |
| Event 7 | 67° | 8 | False | 20180915H00Group27 | 23-09-16.999 |
| Event 8 | 67° | 11 | False | 20180915H00Group52 | 01-05-40.999 |
| Event 9 | 77° | 11 | False | 20180915H00Group32 | 23-30-54.999 |
| Event 10 | 76° | 11 | False | 20180915H00Group49 | 00-51-30.999 |
| Event 11 | 61° | 17 | False | 20180915H06Group53 | 06-22-39.999 |
| Event 12 | 66° | 20 | False | 20180915H06Group78 | 07-46-53.999 |
| Event 13 | 73° | 9 | False | 20180915H12Group25 | 10-51-09.999 |
| Event 14 | 59° | 8 | False | 20180915H18Group30 | 16-39-04.999 |

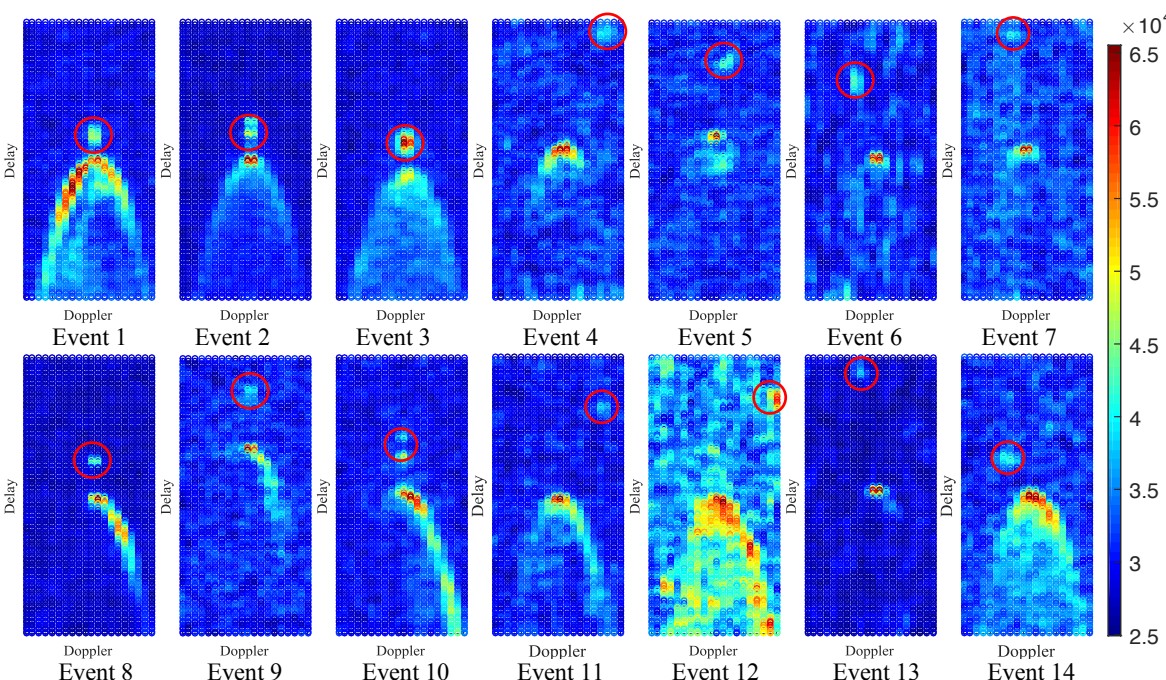

**Figure 2.** Fourteen delay Doppler maps (DDMs) of TDS-1 containing anomalous features that are marked by red circles.

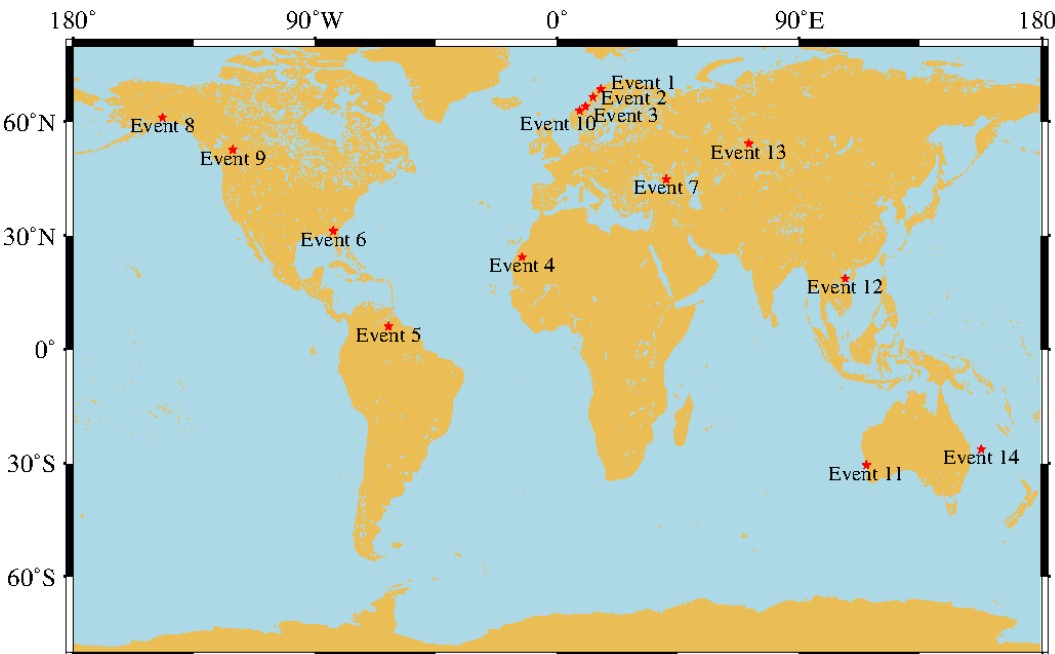

**Figure 3.** The distribution of specular points of the 14 events. The specular points are marked by red stars.

## 3. Analyses of Possible Causes

The generation of DDMs involves three parts from the receiver to transmitter, such as receiver hardware, signal processing, and signal propagation. For typical DDMs the bright points circled in Figure 2 are unexpected. The problems in each part of DDMs generation may cause the bright points. Seven possible causes of the bright points are considered in this work. Three of them relate to receiver hardware, such as (1) random noise, (2) DDM generator losing tracking, and (3) leakage of direct signals. The possible cause relating to the signal processing is aliasing. The remaining three causes are associated with the signal propagation, such as (1) reflections from an Earth's surface

target, (2) GNSS radio-occultation, and (3) reflection from targets above the Earth's surface and under the nadir-looking antenna of the receiver. In the following sections, the first six possible causes are investigated, and evidence found for these not being the most likely causes.

### 3.1. Random Noise

The blue background above the "horseshoe" pattern is utilised to investigate the noise properties for each event. Then a 99% confidence interval is given based on the property information of the noises, which is further used to decide if the bright points are random noise. The bright points of Figure 2 are excluded in noise samples, and there are about 1000 noise samples for each event. This investigation is carried out as follows. First the probability distribution function, mean and standard deviation of the noise samples are obtained. Then the noise samples are assumed to obey normal distribution considering the probability distribution function of the noise samples, and the mean and standard deviation of the noise samples are used to generate a normal distribution function. The next step computes the 99% confidence interval $(a, b)$ using the normal distribution function obtained in the last step. The final step is to determine if the correlation power value of the bright point ($Bval$) is located in the 99% confidence interval $(a, b)$.

Figure 4 gives the probability distribution function for each event, as shown by black dots. The 99% confidence interval $(a, b)$ is indicated by red lines. Since the volume of the noise samples is small and the noise samples may not exactly obey normal distribution, the normal distribution function determined by the mean and standard deviation of the noise samples can not fit the black dots well. Hence the normal distribution functions are normalised to fit the black dots, which are represented by the blue lines of Figure 4. The values for $a, b$ and $Bval$ are given in the top of each sub-plot. It can be seen that the values for $Bval$ are far outside the 99% confidence interval. Therefore, the bright points are not random noises.

### 3.2. DDM Generator Losing Tracking

The investigation of DDM generator is to check if the 14 events happened just after the DDM generator recovers from loss of track. Differential operation is applied to DDMs observation time for the tracks involved in the 14 events to find if and when DDM generator lost tracking. Then the observation time of each event is compared with the DDM generation recovery time. Figure 5 is the differential results of the DDMs observation time of Track ID "20180914H18Group05". This track involves Events 1, 2, and 3, which can be seen from Table 1. The horizontal axis is DDM index in the track, which is of convenience to indicate the observation time. TDS-1 outputs DDMs every second, so the differential results of Figure 5 are mostly 1. However, there are seven outliers which means that the DDM generator loses tracking for seven times during this track. The DDM indexes corresponding to the outliers are shown in the figure. The DDM indexes for Event 1, 2 and 3 are 407, 447, and 498 respectively, which are not around the time that the DDM generator loses tracking. The same results are obtained for the other 11 events, and details are not shown here.

Figure 6 gives the two DDMs that are just before and after the DDM generator loses tracking at DDM index 213 of Figure 5. The DDM indexes are marked by blue rectangles, and observation time by red rectangles. It can be seen from Figure 5 that the DDM generator loses tracking for seven seconds at DDM index 213, which is confirmed by the observation time difference between 15:16:5.999 and 15:16:12.999 of Figure 6. It can be seen from Figure 6 that the two DDMs show a very clear "horseshoe" pattern without any strange patterns. This is understandable because TDS-1 receiver uses on-board-calculated specular point to estimate the delay and Doppler of the reflected signals, to keep tracking the reflected signals. All the tracks relating to the 14 events are checked and it is found that there are no outliers in the positions of the on-board-calculated specular points, even if the DDM generator loses tracking. Therefore, the bright points are not caused by the DDM generator losing tracking.

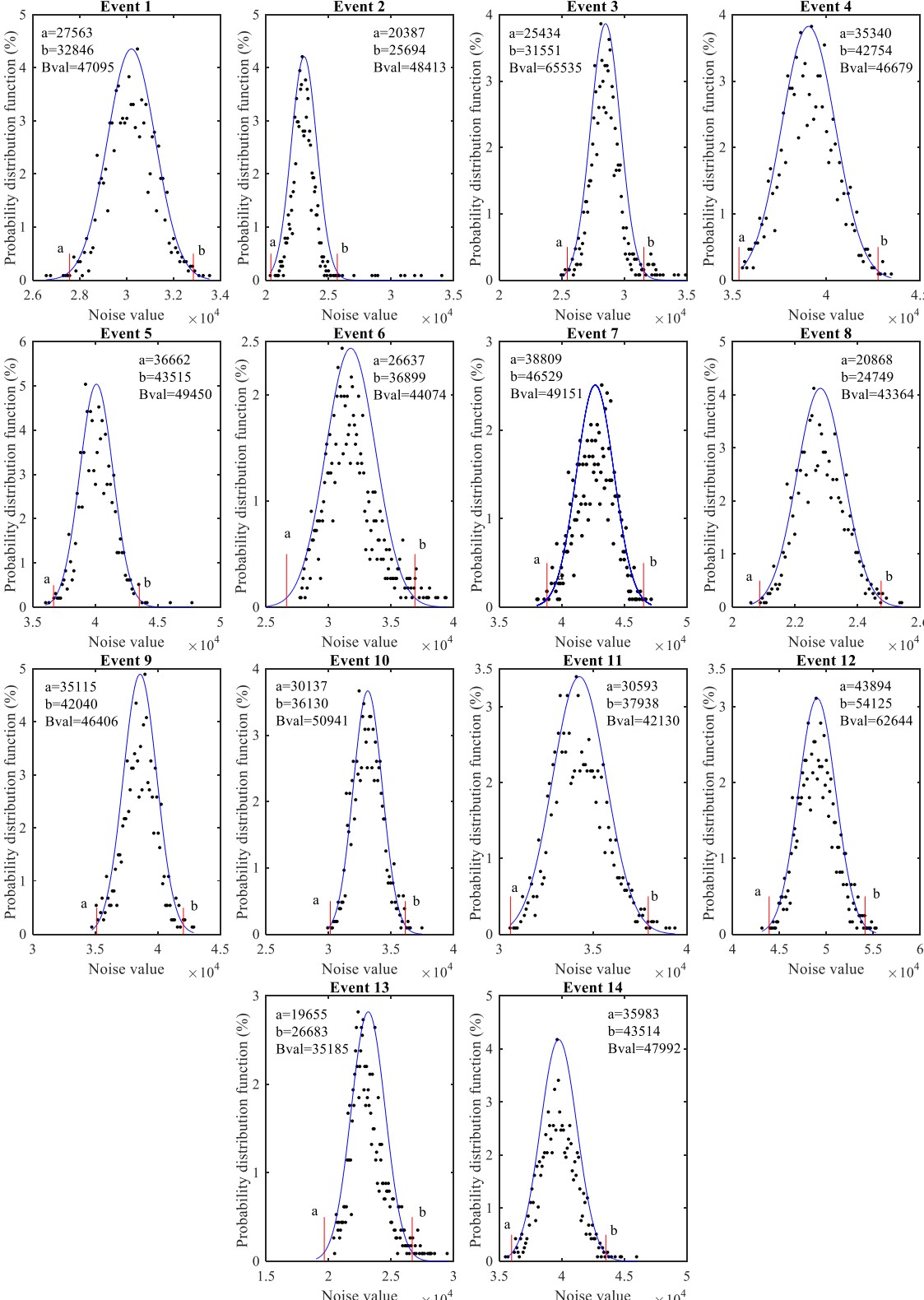

**Figure 4.** Probability distribution functions of the background noises for the 14 events, which are presented by black dots. The noises are assumed to obey normal distributions, and the normal distribution functions with the means and standard deviations corresponding to these of the noises are obtained. $(a, b)$ is 99% confidence interval of the normal distribution functions. The blue curves are the results of normalising the normal distribution functions to fit the black dots. *Bval* is the correlation power value for the bright point.

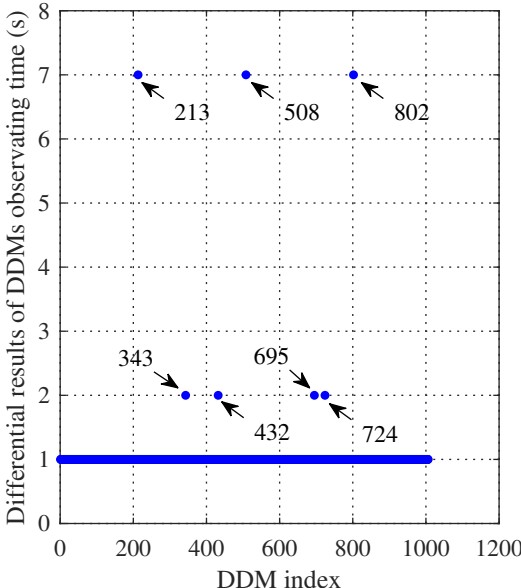

**Figure 5.** Differential results of DDMs observation time of Track ID "20180914H18Group05". The horizontal axis is DDM index, and the DDM indexes corresponding to the seven outliers are indicated by the arrows around them.

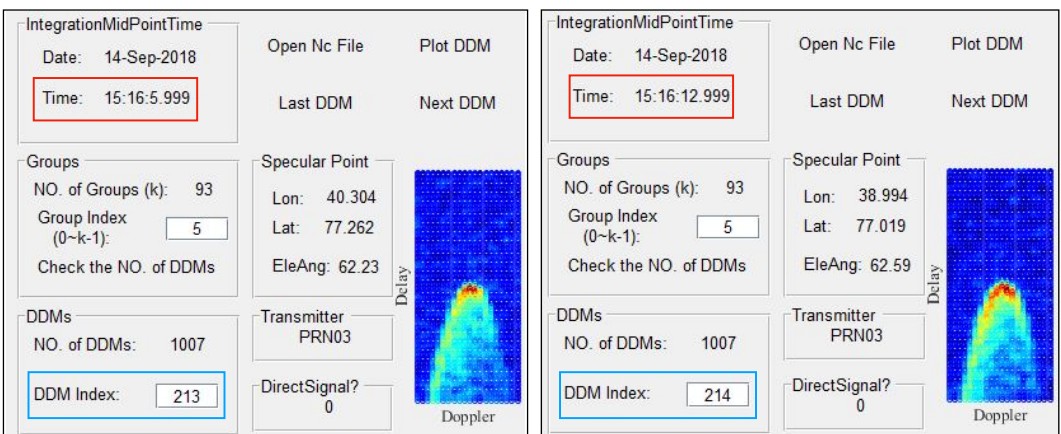

**Figure 6.** Two DDMs that are just before and after DDM generator loses tracking at DDM index 213 of Figure 5.

### 3.3. Leakage of Direct Signal

Two aspects are examined to decide if the bright points correspond to a leakage of the direct signal. One is to analyse the path length and Doppler of the direct signal, and another is to check the system parameter "*DirectSignalInDDM*" as listed in the fourth column of Table 1.

The path length and Doppler frequency difference between the bright points and reflected signals can be extracted from the corresponding DDMs, while the path length and Doppler frequency differences between direct and reflected signals are not directly available from DDMs and other data products. Simulations have been conducted to generate the path length and Doppler frequency of the direct and reflected signals. The simulations use TDS-1 orbit, precise orbit of transmitter, WGS-84 ellipsoid model, corrections for the ionospheric and tropospheric delays, and correction for the mean sea surface height using the DTU13 model [35]. Two parameters, $\rho_{rem}$ for the path length difference between direct and reflected signals and $D_d^S$ for the Doppler frequency difference between direct and reflected signals, are obtained from the simulations:

$$\rho_{\text{rem}} = L_{sp} - L_d - \mu \cdot \tau \cdot c \tag{1}$$

$$D_d^S = D_{dir}^S - D_{ref}^S \tag{2}$$

where $L_{sp}$ is the reflected path length through the specular point, $L_d$ is the path length of the direct signal, $\tau$ is the time period of the C/A code (1 ms), $\mu$ is an integer ambiguity to ensure that $\rho_{rem}$ has a value between 0 and $\tau \cdot c \approx 300$ km, $D_{dir}^S$ and $D_{ref}^S$ are the simulated Doppler frequencies of the direct and reflected signals, respectively. The superscript $S$ of (2) indicates simulated variables, which is used to separate them from observation variables which are marked by the superscript $O$, such as the $D_d^O$ of (4).

The upper delay window of the DDM is defined as:

$$L_{sp} - L_x < Dp \cdot Dr \cdot c \cdot \eta \tag{3}$$

where $L_x$ is the reflected path length from any point, $Dp$ is the number of delay pixels (128 for TDS-1 DDMs), $Dr$ is the delay resolution of DDM (244 ns for TDS-1 DDMs), $c$ is the speed of light, and $\eta$ is the ratio of the number of pixels above the specular point to the number of delay pixels. The upper delay window is the necessary condition that a reflected signal is above the "horseshoe". Given that the specular point is basically located in the mid delay axis of the 14 DDMs, $\eta$ is set to 0.5. Therefore the upper delay window is less than 4682 m, e.g., $0 < L_{sp} - L_x < 4682$ m, and the path length difference between the bright points and reflected signals follows this relation. The Doppler frequency difference between the bright points and reflected signals $D_d^O$ can be obtained from the DDMs [34]:

$$D_d^O = (N - 11) \cdot Dre \tag{4}$$

where $N$ is the column number of the bright point, and $Dre$ is the Doppler resolution (500 Hz in TDS-1). The column numbers of the bright points are listed in Table 1. The specular reflection (DDM peak) is set to be in the 11th column in the TDS-1 DDM.

Table 2 lists the values of $\rho_{rem}$, $D_{ref}^S$, $D_d^S$ and $D_d^O$ for the 14 DDMs. It can be seen that the $\rho_{rem}$ of the DDMs ranges from 15.1 km to 292.6 km, which are considerably beyond the upper delay window. Therefore, the direct signals cannot be above the "horseshoe" of the DDMs, despite the fact that some DDMs have small difference between $D_d^S$ and $D_d^O$ (Events 3, 7 and 13). Furthermore, the path difference between the direct and reflected signals seems to be random. Therefore, the bright points in Figure 2 do not appear to be the leakages of direct signals.

**Table 2.** The values of $\rho_{rem}$, $D_{ref}^S$, $D_d^S$ and $D_d^O$ for the 14 DDMs.

|  | $\rho_{rem}$ (km) | $D_{ref}^S$ (Hz) | $D_d^S$ (Hz) | $D_d^O$ (Hz) |
|---|---|---|---|---|
| Event 1 | 273.1 | 6703 | 1701 | 0 |
| Event 2 | 284.4 | 4726 | 1177 | 0 |
| Event 3 | 292.6 | 2242 | 518 | 0 |
| Event 4 | 186.7 | 2081 | 453 | 3000 |
| Event 5 | 54.7 | −3114 | −745 | 500 |
| Event 6 | 116.3 | 15812 | 3826 | −1500 |
| Event 7 | 221.1 | −6864 | −1595 | −1500 |
| Event 8 | 224.1 | 5715 | 1418 | 0 |
| Event 9 | 22.4 | −5668 | −1622 | 0 |
| Event 10 | 15.1 | −8527 | −2079 | 0 |
| Event 11 | 156.7 | 2087 | 730 | 3000 |
| Event 12 | 211.8 | −14858 | −3780 | 4500 |
| Event 13 | 290.8 | −3627 | −1052 | −1000 |
| Event 14 | 119.4 | −80 | −173 | −1500 |

The system parameter "*DirectSignalInDDM*" indicates if the reflected signals are separated from direct signals. It did not catch all cases in early version data set, but the problem was solved in data products after February 2018 [34]. It can be seen from Table 1 that the "*DirectSignalInDDM*" of the 14

DDMs are "False", which means that the DDMs do not contain any direct signal, which confirms the previous analysis.

### 3.4. Aliasing

Aliasing is the signal ambiguity in the frequency domain [36]. An appropriate sampling frequency could avoid this issue according to the Nyquist-Shannon sampling theorem. In TDS-1 signal processing, two processes involve sampling [34]. One is the sampling of the centre frequency signal (4.188 MHz) using a sampling frequency of 16.367 MHz. The other one is the decimation which makes the effective sampling frequency drop to 32 kHz. More details about the two processes can be found in [34]. It is known that DDMs are the correlation results in two domains (time and frequency), which means that the bright points or the "horseshoe" of Figure 2 are produced with appropriate delay and Doppler frequency. Given that aliasing is just the ambiguity in the frequency domain, it cannot produce the bright points without appropriate delays. Therefore, the bright points of Figure 2 do not appear to be due to aliasing.

### 3.5. Reflections from an Earth's Surface Target

It can be seen from Figure 2 that the bright points are located in delay bins before the "horseshoe" pattern, i.e., before the specular reflection from the Earth's surface arrives. These regions are often called the "forbidden zone" of the DDM because they correspond to delays shorter than the delay associated with the points over the Earth's surface. Therefore, the bright points of Figure 2 cannot be caused by reflections from an Earth's surface target. Moreover, the height difference corresponding to the path length difference between the bright point and specular point is obtained to investigate if the bright point is caused by properties on the Earth's surface. The height difference can be obtained by [12]

$$\Delta h = -\frac{\Delta \rho}{2 \sin \Theta} \tag{5}$$

where $\Delta \rho$ is the path length difference between the bright point and specular point, and $\Theta$ the elevation angle which is given in Table 1. The negative sign means that height measurements decrease with the path lengths of reflected signals. The C/A code integer ambiguity $\mu$ in (1) is set to zero in this investigation to obtain the minimum value of the height difference for each event. It is obtained that the height differences for the 14 events are greater than 445 m (Event 1), and can reach to thousands of metres such as Event 7. Skyscrapers are the properties of which the heights are hundreds of metres, but it is found that the reflection areas for the 14 events have no skyscrapers that match the corresponding height differences. Hence the bright points of Figure 2 are not caused by the properties on the Earth's surface.

### 3.6. GNSS Radio-Occultation

GNSS radio occultation (GNSS-RO) event happens when GNSS satellites are setting or rising behind the Earth's limb as viewed by a receiver fixed on LEO satellite [37], which is shown by the red line of Figure 7. In addition, Figure 7 gives an illustration of GNSS-R which is presented by black lines. The GNSS-RO uses refracted signals, while the GNSS-R uses reflected signals. It can imagine that the GNSS-R may be close to the GNSS-RO when the elevation angle is very low, such as about $0°$. However, in the case of the 14 events, their elevation angles which are listed in Table 1 are considerably high, ranging from $59°$ to $85°$ which precludes the occurrence of GNSS-RO events.

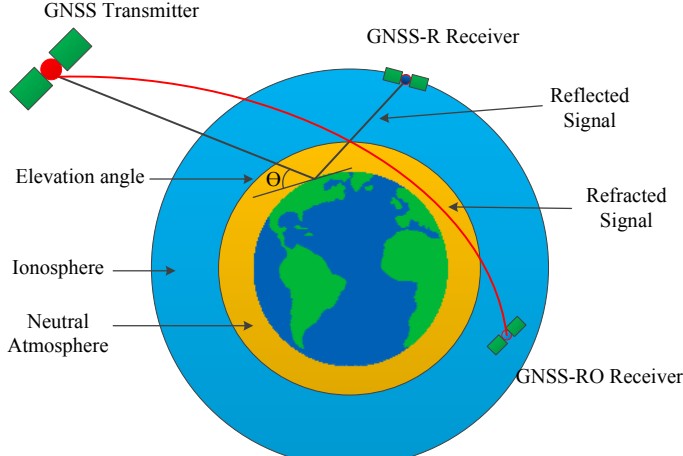

**Figure 7.** Illustration GNSS-R and GNSS-RO, which are shown by black and red lines respectively. $\Theta$ is elevation angle.

## 4. Object Positioning

The bright points of Figure 2 appear to be the reflection from objects above the Earth's surface because their path lengths are shorter than those of the specular points. This section describes the method used to calculate their position using the delay, Doppler frequency and appropriate geometry, and gives the conditions that the suspected objects have to fulfil.

The retrieval of target position using delay and Doppler frequency usually has no unique solution even if the target is on the Earth's surface. This issue is more complicated for space target reflections. Figure 8 gives an illustration of the geometry for the retrieval of object position which are shown by black lines. The solid lines represent direct signals, and the dashed lines the reflected signals. This geometry has two main assumptions, (1) the altitude of the target is lower than that of the receiver, and (2) the normal vector of the reflecting surface is perpendicular to the Earth's surface. The Earth's surface is assumed to be the WGS-84 ellipsoid model. It can be seen from Figure 8 that the target could be at different altitudes due to the integer ambiguity $\mu$ of (1). It is possible that the reflection could happen under other conditions, with a typical example shown in Figure 8 by blue lines. Since other reflection conditions are unpredictable, they are not considered in the retrieval of target position.

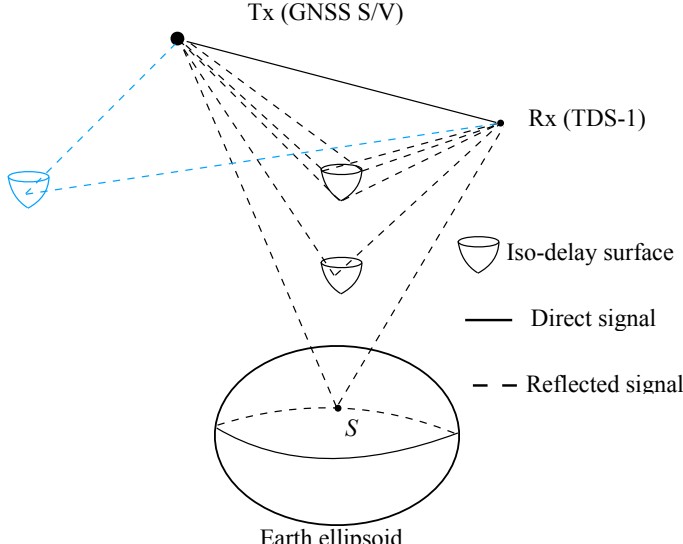

**Figure 8.** Illustration of geometry model used in the retrieval of the target position, as shown by black lines. The blue lines give a typical example of unpredictable reflection conditions. *S* is specular point on the Earth's surface.

Three steps are used to retrieve the target positions. The first step obtains the path lengths and Doppler frequency of the bright points. The delay and Doppler frequency of the TDS-1 DDMs are relative to those of the on-board-calculated specular point to keep the specular reflection in the middle of the DDM. The Doppler frequency pixel corresponding to the on-board specular point is the 11th column, which is set to zero, and the delay pixel corresponding to the on-board specular point can be obtained using the system parameters "*TrackingOffsetDelayNs*" and "*SpecularPathRangeOffset*" [34]. The delay and Doppler frequency of the bright point relative to those of the on-board specular point can be estimated from the number of pixels between them. The TDS-1 dataset does not provide the path length and Doppler frequency of the on-board specular point directly. The path length of the on-board specular point is obtained using the on-board receiver position, on-board transmitter position, and on-board specular point position. Similarly the Doppler frequency of the on-board specular point is simulated using on-board trajectory information of the receiver, transmitter, and the position of on-board specular point.

The second step is positioning the target using path length information. It is known that a given path delay (except for specular delay) corresponds to a branch of reflection points, which form an iso-delay curve in the two-dimensional domain and an iso-delay surface in the three-dimensional domain. The iso-delay surface is shaped like a "top", as shown in Figure 9. In this investigation the "top" is sliced into 11 iso-delay curves, and the specular point of each iso-delay curve is marked by blue dots in Figure 9. The path length difference of the specular points of two adjacent iso-delay curves is 30 m. Considering the C/A code integer ambiguity $\mu$, the possible positions of the target will have several iso-delay surfaces in one event.

The last step is based on the results of the second one, and uses Doppler frequency information to further constrain the target position. Two assumptions are made on the motion of the target: (1) the velocity of the target is perpendicular to the radial vector of the target, and (2) the target describes a nearly circular orbit when $\mu > 0$, and the speed $v$ is determined from Kepler's equation. The target is assumed to be static when $\mu = 0$. These assumptions are reasonable for satellite orbits with very small eccentricities, but have limitations when the eccentricities are too large to produce nearly circular orbits. In addition, the assumptions do not consider the situations that the targets are moving when $\mu = 0$ because in this case the moving direction is arbitrary. Hence the assumptions work in the normal reflection as shown by black lines of Figure 8. For a given target position such as the red dot of Figure 9, the Doppler frequency corresponding to each moving direction is calculated, and compared with the Doppler frequency from the first step. If the Doppler frequencies of all directions are considerably different from the Doppler frequency of the first step, the given position will be eliminated.

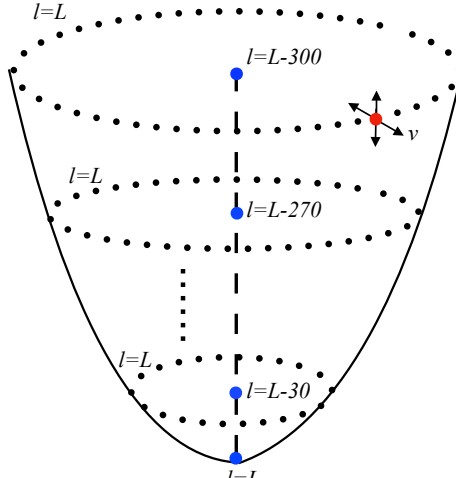

**Figure 9.** Illustration of iso-delay surface in three-dimensional domain. *l* indicates path length, and *L* is a given path length. The black dots are the possible positions of the target, and the blue dots are the specular points of the corresponding iso-delay curve.

These three conditions (delay, Doppler and geometry) are considered in the verification of a target candidate. The path delay $d^T$, and Doppler frequency $D^T$ of a given target candidate can be calculated using the position and velocity information of the transmitter, receiver, and the target candidate. In addition, the path delay $d^O$, and Doppler frequency $D^O$ of the bright points of Figure 2 were already obtained in the previous part. Then, the path delay difference $d^{OT}$ between a particular target candidate and the bright point can be obtained, and similarly, the Doppler frequency difference $D^{OT}$ can also be obtained. It should be noted that $d^{OT}$ is between 0 and $\tau \cdot c \approx 300$ km, which eliminates the impact of C/A code ambiguity. The delay and Doppler frequency conditions are that $d^{OT}$ and $D^{OT}$ should be smaller than particular thresholds, which are discussed in the next section. The geometry condition is that the target candidate can be seen by the transmitter, and that its altitude is less than that of the TDS-1.

## 5. Results Analyses

### 5.1. Target Position Retrieval

Figure 10 gives the results of target positions of Event 11, with four sub-plots for C/A code ambiguity $\mu = 0$, 1, 2 and 3. The blue dots are the results considering only the delay, and the red dots considering both the delay and Doppler frequency. The dots show a "top" pattern in all sub-plots, which confirms the analyses of Figure 9. It can be seen that in sub-plot (a) where the "target" is near the Earth's surface ($\mu = 0$), there are no red dots, which means there are no solutions under the geometry assumption of Figure 9 and the assumption that the "target" is static. This is understandable because the Doppler frequency difference between the bright point and specular point of Event 11 is 3000 Hz according to Table 2, which is large because the Doppler frequency of a scattering point of which path delay is less than 300 m is usually less than 1500 Hz as compared to that of a specular point on the Earth's surface. When $\mu > 0$, there are an increasing number of red dots as shown in sub-plots (b) to (d). The results for target positions of other events are similar to Figure 10, which are not presented here.

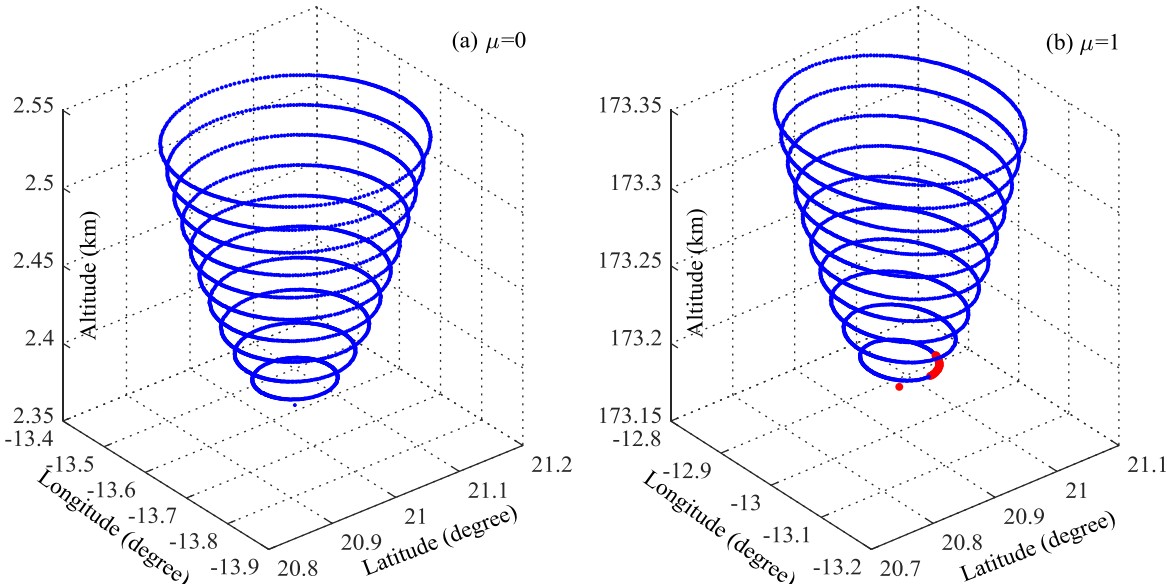

**Figure 10.** *Cont.*

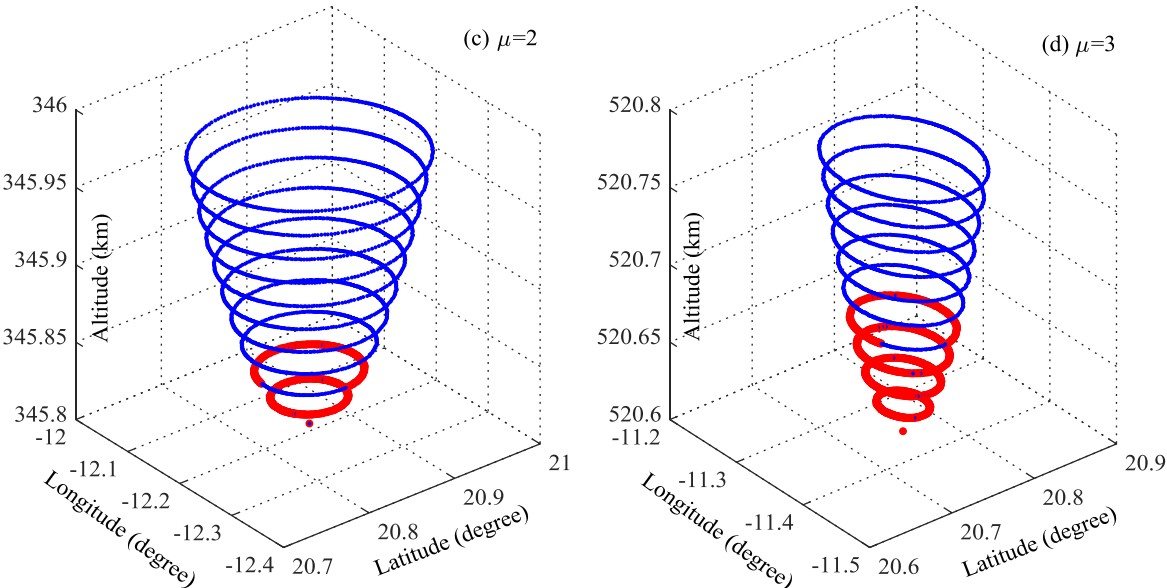

**Figure 10.** Results of target positions of Event 11. Sub-plots (**a**) and (**b**) correspond to $\mu = 0$ and $\mu = 1$, respectively. Sub-plots (**c**) and (**d**) correspond to $\mu = 2$ and $\mu = 3$, respectively. The blue dots are the results only considering the delay condition, and the red dots are the results further considering the Doppler frequency condition.

*5.2. Verification of Suspected Objects*

The UCS provides the list of operational satellites orbiting the Earth, and updates the database regularly. The database used in this investigation was issued on 1 December 2018. There are 1957 satellites in the database, of which 834 remain after applying such filter conditions as: (1) the class of orbit is LEO, (2) perigee is less than 650 km considering that TDS-1 altitude is around 640 km, and (3) launch date is before 15 September 2018. Other objects such as debris and aircraft are not considered in this investigation, which is explained Section 6.

The position and velocity of the 834 satellites are obtained by simplified general perturbations 4 (SGP4) method [38] using two line elements (TLE) [39]. The TLE information can be accessed from Space-Track [39]. The TLE of 746 satellites at the time around the observing period of the 14 DDMs were obtained from the Space-Track, while the other 85 satellites were not available. Therefore, 746 satellites are the target candidates, and the $d^{OT}$ and $D^{OT}$ were obtained by calculation. The position precision of SGP4 is around 2.5 km for the TDS-1, by comparison with the TDS-1 on-board position, which could be a reference for the target candidates.

The threshold value of $d^{OT}$ is set to 10 km considering the SGP4 orbit error, TLE age, and other error sources including path delay error caused by the DDMs, and orbit errors of the transmitter and receiver. The threshold value of $D^{OT}$ is set to around 1000 Hz considering the DDM Doppler resolution is 500 Hz. Three events have valid candidates after applying the criteria that $d^{OT} < 10$ km, $D^{OT} < or \approx 1000$ Hz, and the geometry condition. The relevant information of the three events is listed in Table 3, including $d^{OT}$, $D^{OT}$, and the altitude of the target candidate. It can be seen from Table 3 that the valid candidates for Events 2, 4 and 5 are Kondor, Step Cube Lab and Deimos 2, respectively. The values of $d^{OT}$ for Kondor, Step Cube Lab and Deimos 2 are around 500 m which is small. This is understandable because (1) the reflected path length is usually insensitive to the position of scattering point because the semi-major and semi-minor of an iso-delay curve is from several kilometres to tens of kilometres in space-based scenario [40], (2) the path delay extracted from the DDMs is of the order of tens of metres, which is related to the delay resolution (244 ns), and (3) the errors induced by other factors such as the orbit of the transmitter and receiver are much smaller, and considered to be negligible. In addition, the $D^{OT}$ of the three candidates are 607 Hz, 568 Hz

and 1046 Hz respectively, which are also reasonable considering the Doppler resolution of 500 Hz for the TDS-1 DDMs. Therefore, Kondor, Step Cube Lab and Deimos 2 can be said to meet the delay and Doppler frequency conditions, so they are possibly targets of the bright points for Events 2, 4 and 5, respectively.

**Table 3.** Three events with valid target candidates.

|  | Name and NORAD ID | $d^{OT}$ (km) | $D^{OT}$ (Hz) | Altitude (km) |
|---|---|---|---|---|
| Event 2 | Kondor (39194) | 0.495 | 607 | 459.5 |
| Event 4 | Step Cube Lab (43138) | 0.458 | 568 | 505.9 |
| Event 5 | Deimos 2 (40013) | 0.653 | 1046 | 616.4 |

In addition, the sub-satellite positions of the transmitter, receiver and valid target candidates of the three events are shown in Figure 11. The upper, middle and lower sub-figures are for the Event 2, 4 and 5 respectively. In normal reflection of GNSS-R, such as the dashed line through *S* in Figure 8, the sub-satellite positions of transmitter, receiver and reflection point (*S*) are on a line. It can be seen from Figure 11 that the sub-satellite point of the target candidate is far away from the line formed by those of the transmitter and receiver for the three events. This means the reflection conditions of the three events are like the one shown by the blue lines of Figure 8.

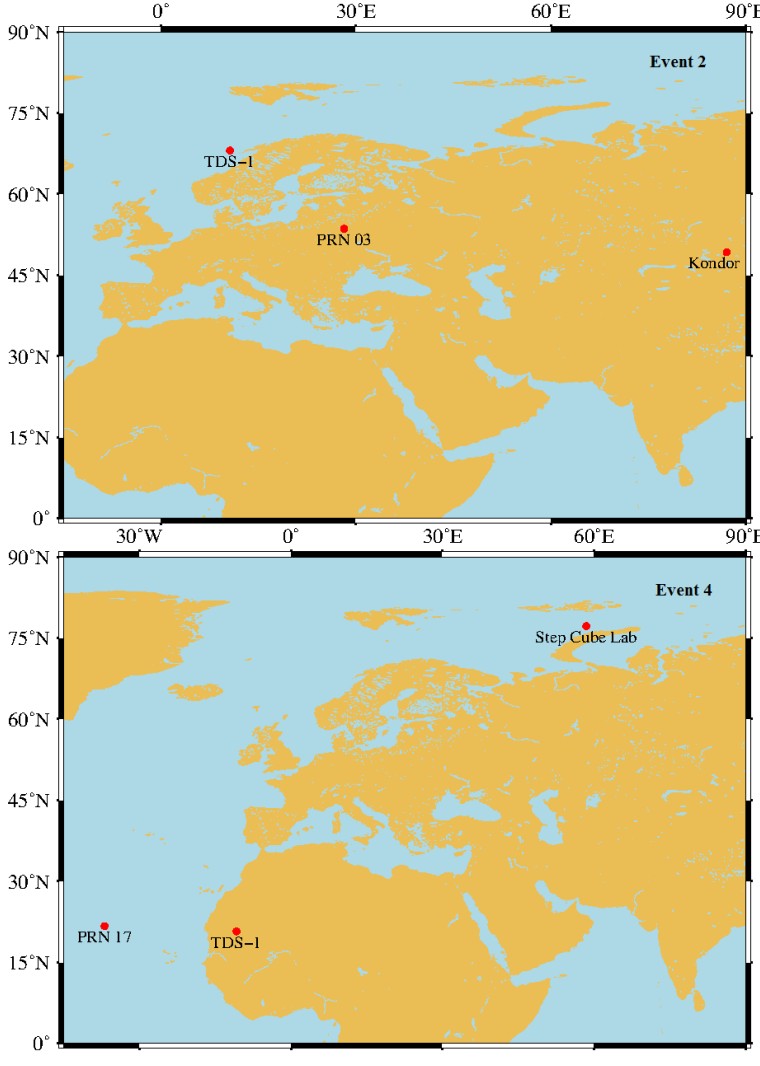

**Figure 11.** *Cont.*

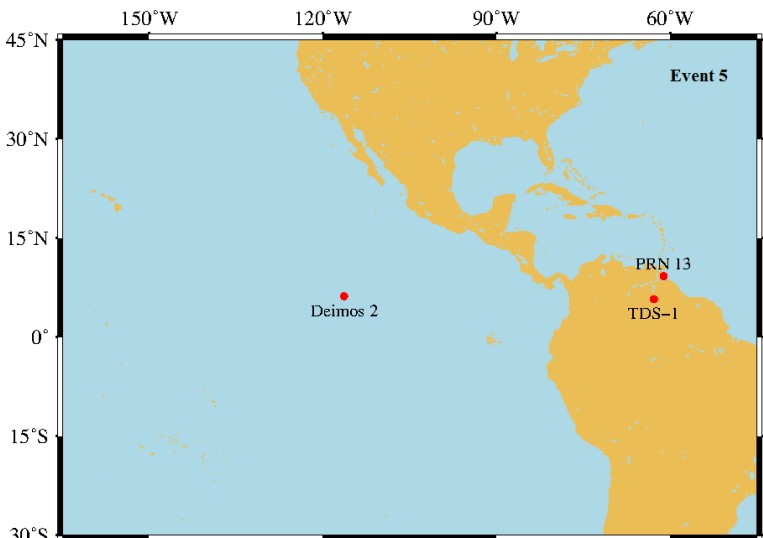

**Figure 11.** The sub-satellite locations of transmitter, receiver and valid target candidates for Event 2 (upper figure), 4 (middle figure), and 5 (lower figure), which are shown by red dots.

## 6. Conclusions and Remarks

This work proposed a new application of GNSS-R to detect objects above the Earth's surface such as satellites. This finding was derived from the analysis of 14 DDMs of the TDS-1 mission, which contain anomalous reflections, as the bright points (Figure 2) occurring in the "forbidden" zone of the DDMs. Seven possible causes of the unusually bright points were analysed, including random noise, DDM generator losing tracking, leakage of direct signals, aliasing, reflection from an Earth's surface target, GNSS radio-occultation, and reflections from objects above the Earth's surface. The conclusion was that the bright points are likely coming from the reflection of targets above the Earth's surface according to their delays. Two parts of work were done based on this conclusion, target positioning by theoretical analyses and target search in satellite database. It is challenging to theoretically retrieve the target positions because there are unpredictable reflections, such as the reflection shown by the blues lines in Figure 8. Hence this work investigated the target positions under normal reflection conditions (black lines of Figure 8). Moreover, the UCS satellite database has been searched, and it was found that Kondor, Step Cube Lab, and Deimos 2 may be responsible for Events 2, 4 and 5, as they meet the delay and Doppler frequency conditions, with delay error $d^{OT}$ around 500 m and Doppler error $D^{OT}$ less than 1050 Hz.

These 14 DDMs containing anomalies were found in a time period of less than two days, which indicates that these anomalies are not rare. If there were no other reasons for the anomalies, GNSS-R will be possible to detect targets above the Earth's surface. This work is a preliminary study of this new application, and there are two limitations which leave space for future work. First, the location distributions of the 14 DDMs are in land or coastal areas, and none of them is in open ocean. This is due to the limited number of events, if more events could be found, a random distribution pattern would be seen. The future work will use both TDS-1 and CYGNSS dataset to find more events. Second, three events have valid target candidates, while no candidates are found for the other 11 events. In the search for target candidates, one satellite database was used, and other databases such as aircraft and debris are not included because it is challenging to gather the database which is effective in the observing period, and that provides accurate position information. The future work will gather a more effective database to search the target candidates.

**Author Contributions:** Conceptualization, C.H. and C.B.; methodology, C.H.; formal analysis, C.H., C.B., H.P. and A.C.; resources, C.H.; writing—original draft preparation, C.H.; writing—review, C.B., A.C., L.Q. and C.R.; supervision, C.B., L.Q. and C.R.

**Funding:** This research received no external funding.

**Conflicts of Interest:** The authors declare no conflict of interest.

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
