# Peer review of "Detecting Targets above the Earth’s Surface Using GNSS-R Delay Doppler Maps: Results from TDS-1"

_remotesensing, doi:10.3390/rs11192327_

Round 1

Reviewer 1 Report

This paper is interesting to present applications of GNSS-R to detect objects above the Earth’s surface from TDS-1. However, it cannot guarantee that the anomalous reflections, as the bright points occurring in the “forbidden" zone of the DDMs are caused by the Earth’s surface target, or any other properties land or ocean surfaces, or other noises. Of course any DDMs can retrieve Earth's surface positions under normal reflection conditions, but it is difficult to confirm Kondor, Step Cube Lab, and Deimos 2 that are responsible for Events 2, 4 and 5. It needs more data to check and validate it.

Therefore, it cannot recommend publication in the current form.

Reviewer 2 Report

Please find my comments attached.

Reviewer 3 Report

The authors present a method to implement GNSS reflection to detect the objects above the Earth’s surface, which is novel and worth to publish after minor correction. The detailed comments include:

1. The second paragraph of the introduction section. In the literature review, the authors talk about some results have been achieved (like “good results have been achieved” “experimental results have been reported”). The results are expected to provide with more details and some comments.

2. Page 2, second paragraph. Give some comments on why GNSS-R technique to detect objects is useful and how it is beyond the current methods.

3. Page 3, first paragraph, the last sentence. Some theoretical support (some references for example) should be added to justify your statement.

4. Page 4, first paragraph of Section 3. How are the five points proposed? This should be explained.

5. Page 5, last sentence of Section 3.2. More evidence or analysis should be added to support the statement.

6. Equation 2, the definition of superscript s is missing.

7. Page 7, first paragraph, “Since other reflection conditions are unpredictable, …”. What are the other reflection conditions? Please give some representative examples. How would they influence the results? You should let the readers know the advantages and limitations of your method.

8. Page 7, the last paragraph. Give some comments on the assumptions. In what conditions, they would not hold? How would the assumptions influence the performance?

9.Page 11, the conclusion. The sentence from “Then, the retrieved possible target positions under normal reflection conditions …”. As part of the conclusion, the results and conclusions from that are not clear. More like findings not conclusion. Suggest to reword it.

Round 2

Reviewer 2 Report

Overall the paper has improved considerably. Authors discussed two extra cases of potential origin of the observed anomaly: lost of synchronise of the DD generator and random noise.

This reviewer is  now convinced that the anomaly is not noise but not sure that follows the explanation provided by the authors, for instance what is Bval? Clarify, not sure why authors assumed normal distribution, could you provide a reference for that? Usually power of noise is Raylegh instead of Normal.

Have authors considered that the loose tracking might happen inside the one second integration time? Please discuss it in the manuscript

Why this effect has not been observed by any other GNSS-R mission, what makes TDS-1 special that allows this measurement. Please discuss it in the manuscript.

The anomaly shape is almost always the same: it spreads two adjacent Doppler lags and some delay  lags. And in most of the cases (Evet 1, 2, 3, 6, 8, 9 and 10) it is just some delays away from minimum Doppler/DDM peak. How this matches with the fact that this is a satellite reflection? I am aware that authors have already discussed the possibility of a hardware error per my request. But the shape is too systematic, t always looks the same, too much coincidence. My guess is this lack of synch within 1 second incoherent accumulation.

Can you please provide a colobar unified for all the DDM in Fig 2? For instance Event 12 is a really weak signal that the anomaly could well be noise.

line 119 -> "the noise samples are assumed to obey normal distribution" please provide a reference for that. line 149 -> "becuase"

I see that the conclusion is cautious, as it should be. I even suggest saying something like "a satellite is a plausible explanation for the anomaly although for being certain we need to analyse further data ". In contradiction with what I said in the first review, maybe it would be a good idea to reflect some of this uncertainty on the manuscript title.
